# Structures, Antioxidant Properties, and Antimicrobial Properties of Eu(III), Gd(III), and Dy(III) Caffeinates and *p*-Coumarates

**DOI:** 10.3390/molecules28186506

**Published:** 2023-09-07

**Authors:** Grzegorz Świderski, Monika Kalinowska, Ewelina Gołębiewska, Renata Świsłocka, Włodzimierz Lewandowski, Natalia Kowalczyk, Monika Naumowicz, Adam Cudowski, Anna Pietryczuk, Edyta Nalewajko-Sieliwoniuk, Izabela Wysocka, Żaneta Arciszewska, Beata Godlewska-Żyłkiewicz

**Affiliations:** 1Department of Chemistry Biology and Biotechnology, Bialystok University of Technology, Wiejska 45E, 15-351 Bialystok, Polande.golebiewska@pb.edu.pl (E.G.); n.kow.1347@gmail.com (N.K.); 2Department of Physical Chemistry, Faculty of Chemistry, University of Bialystok, K. Ciołkowskiego 1K, 15-245 Białystok, Poland; 3Department of Water Ecology, Faculty of Biology, University of Bialystok, Ciołkowskiego 1J, 15-245 Bialystok, Poland; cudad@uwb.edu.pl (A.C.); annapiet@uwb.edu.pl (A.P.); 4Department of Analytical Chemistry, Faculty of Chemistry, University of Bialystok, K. Ciołkowskiego 1K, 15-245 Białystok, Polandz.arciszewska@uwb.edu.pl (Ż.A.); bgodlew@uwb.edu.pl (B.G.-Ż.)

**Keywords:** lanthanides, spectroscopic study, antioxidant properties, antimicrobial properties, p-coumarates, caffeinates

## Abstract

In this study, we investigated the structures of lanthanide (Eu(III), Dy(III), and Gd(III)) complexes with *p*-coumaric (p-CAH_2_) and caffeic (CFAH_3_) acids using the FTIR_KBr_, FTIR_ATR_, and Raman spectroscopic methods. The compositions of the solid phase caffeinates and *p*-coumarates were obtained on the basis of the amounts of hydrogen and carbon determined using an elemental analysis. The degree of hydration and the thermal decomposition of each compound were examined via a thermal analysis of TG, DTG, and DSC. Antioxidant spectroscopic tests were performed using the DPPH (1,1-diphenyl-2-picrylhydrazyl radical), FRAP (ferric reducing antioxidant activity), and ABTS (2,2’-azino-bis-(3-ethylbenzothiazoline-6-sulfonic acid) (diammonium salt radical cation) methods. The antimicrobial activity of each compound against *Escherichia coli*, *Bacillus subtilis,* and *Candida albicans* was investigated. The electrical properties of the liposomes which mimicked the microbial surfaces formed in the electrolyte containing the tested compounds were also investigated. The above biological properties of the obtained complexes were compared with the activities of p-CAH_2_ and CFAH_3_. The obtained data suggest that lanthanide complexes are much more thermally stable and have higher antimicrobial and antioxidant properties than the ligands (with the exception of CFAH_3_ in the case of antioxidant activity tests). The Gd(III) complexes revealed the highest biological activity among the studied lanthanide complexes.

## 1. Introduction

Caffeic acid (CFAH_3_) and para-coumaric acid (p-CAH_2_) (Figure 1) are hydroxy derivatives of cinnamic acid and are characterized by a wide range of health-promoting effects [1,2]. They are secondary plant metabolites and are formed from phenylalanine in the pathway of hydroxycinnamic acid synthesis. In fruits, vegetables, cereal grains, wine, and coffee, p-CAH_2_ and CFAH_3_ occur in many forms—monomers, dimers, oligomeric derivatives and glycosides, amides, organic esters, and other chemical compounds [3]. Plant polyphenols, which include CFAH_3_ and p-CAH_2_, have antioxidant and pro-oxidant properties. They can effectively neutralize free radicals, as has been extensively researched and described in the literature [4,5]. On the other hand, the pro-oxidant properties of phenolic compounds may be important in their anticancer activity and in inducing apoptosis [6,7,8]. CFAH_3_, when administered in cancer chemotherapy in combination with drugs such as cisplatin and doxorubicin, may increase the ability of the pharmaceuticals to inactivate cancer cells [9,10]. It has also been shown that CFAH_3_ and p-CAH_2_ have antimicrobial [3,11,12,13,14,15], antidiabetic [16,17], and anti-inflammatory effects [18].

Biological activity, including the antimicrobial, antioxidant, and anticancer activities of organic ligands from the group of phenolic acids, may change after the formation of complexes with metal cations [19,20,21,22]. The reactivity of a chemical compound and its biological effects depend on the molecular structure of the molecule. The activity of a metal complex changes as the metal attached to the ligand changes [23,24,25]. Phenolic compounds not only coordinate metals through the carboxylate group, but they can also attach metals through the hydroxyl groups of the aromatic ring. In phenolic acid complexes in which the metal is attached to the ligand through the hydroxyl groups of the aromatic ring, the antioxidant activity may be reduced compared with that of the uncomplexed ligand. This has been observed, e.g., in the case of CFAH_3_ complexes, whose antiradical activity was found to be lower than that of CFAH_3_ [19]. In complexes of p-CAH_2_ and other phenolic acids in which the metal is attached via the carboxylate group, increases in the antioxidant activity against the ligand have been observed [20,22].

Complexes of natural compounds with lanthanides are interesting chemical compounds in terms of their pharmacological activity. Lanthanide(III) ions have ionic radii similar in size to calcium ions (Ca^2+^), and they may show similarities in biological activity (affinity to the same binding sites in biological systems). Therefore, many studies have focused on discovering their potential medical applications [26,27]. Lanthanide salts are highly toxic, so the possibility of developing a variety of new compounds based on lanthanide ions that would show both therapeutic effectiveness and therapeutic efficacy in cancer treatment while being minimally harmful to healthy cells is still being investigated [28]. In the available literature, a significant part of the research has been devoted to lanthanide complexes with Schiff bases. It has been shown that lanthanide complexes with these ligands may be characterized by higher anticancer activity than the ligands alone [29,30]. Moreover, some lanthanide complexes with organic ligands (derivatives of some phenolic acids) have been found to have higher antimicrobial activity than the initial compounds [31,32]. The presence of lanthanide(III) ions also influences the antioxidant activity of ligands [19]. 

In this work, the structures of solid p-CAH_2_ and CFAH_3_ complexes with Eu(III), Dy(III), and Gd(III) were studied by means of many complementary spectroscopic methods. Different types of metal–ligand coordination, as well as the impact of this type of coordination on the biological activities of the complexes, are discussed. 

## 2. Results and Discussion

### 2.1. Elemental and Thermal Analysis

#### 2.1.1. Elemental Analysis

Table 1 presents the results of the elemental analysis of the Dy(III), Eu(III), and Gd(III) complexes with p-CAH_2_ and CFAH_3_. The results show that in the solid complexes of Dy (III), Eu(III), and Gd(III) with CFAH_3_, the molar metal:ligand ratio was 1:3. In the case of the solid complexes of the investigated lanthanides with p-CAH_2_, the molar metal:ligand ratio was 1:3.

#### 2.1.2. Thermal Analysis

The degree of hydration of the obtained complexes was confirmed by thermal analysis of the tested complexes. Figure 1 and Figure 2 show the thermal decomposition curves (TG, DTG, and DSC) of Eu(III), Gd(III), and Dy(III) complexes with p-CAH_2_ and CFAH_3_, respectively. Table 2 presents the results of the thermal analysis of the tested complexes, i.e., the temperature of the individual stages of decomposition (dehydration, thermal decomposition) and the decomposition products.

CFAH_3_ and p-CAH_2_ are thermally stable compounds up to about 180–200 °C. At the temperature of 180 °C, the process of thermal decomposition of p-CAH_2_ begins. Initially, the decomposition curve changes little, and then, at about 200 °C, the curve jumps sharply (up to 220 °C). The DSC curve indicates the ongoing endothermic process associated with decarboxylation, detachment of the hydroxyl group, and the aliphatic chain from the aromatic ring of the acid, which is indicated by a decrease in weight to about 52%. In the second stage, starting at 220 °C, carbon and hydrogen from the aromatic ring are combusted, and the DSC curve indicates an exothermic process. At a temperature of about 550 °C, a small mass of decomposition products remains, about 2% (probably organic carbon residues), which undergo further slow combustion until the end of the process at 995 °C. As a result of the thermal decomposition of p-CAH_2_ carried out in an oxygen atmosphere, we do not obtain any residues of the products of this process.

Thermal decomposition of CFAH_3_ begins at a temperature of about 200 °C and we observed a rapid decrease in the TG curve to 231 °C. In this stage, the acid is decarboxylated and the hydroxyl group is lost (weight loss of about 30%). A peak related to the endothermic process (223.6 °C) was noted on the DSC curve. In the next stage of thermal decomposition, the aliphatic chain and the aromatic ring of the acid are decomposed. This process lasts up to a temperature of about 577 °C. Above this temperature, there are practically no decomposition products (only the remains of organic carbon, which are burnt to the temperature of 955 °C until the end of the process).

In the process of thermal decomposition of Dy(p-CAH)_3_ and Dy(CFAH_2_)_3_, the first step is the dehydration of the complexes. The Dy(CFAH_2_)_3_ loses six water molecules up to 200 °C, while the Dy(p-CAH)_3_ loses five water molecules up to 180 °C. Two steps can be observed in the dehydration step of Dy(p-CAH)_3_ and Dy(CFAH_2_)_3_. In the first stage, reaching about 100 °C, the complexes lose three molecules of water in Dy(CFAH_2_)_3_ and three molecules of water of crystallization in Dy(p-CAH)3. In the further process of thermal decomposition, the p-coumaric acid complex loses two molecules of water of hydration up to the temperature of about 180 °C and the caffeic acid complex loses three molecules of water. Then, the thermal decomposition of the complexes takes place. On the TG thermal decomposition curve of the Dy(CFAH_2_)_3_, it is impossible to clearly distinguish the individual stages of decomposition (decarboxylation, aromatic ring disintegration). This curve has a slightly different shape than the Dy(p-CAH)_3_ decomposition curve. It is related to different types of metal–ligand coordination in both complexes. The thermal decomposition process of Dy(CFAH_2_)_3_ begins at 200 °C. Around the temperature of 600 °C, stable dysprosium oxide (Dy_2_O_3_) and organic carbon residues remain, which are slowly burnt out. The process of thermal decomposition of Dy(p-CAH)_3_ begins at 180 °C and, similarly to the Dy(p-CAH)_3_, ends with obtaining a stable product of Dy(III) oxide Dy_2_O_3_ at 550 °C. Several spikes are observed in the TG degradation curve of Dy(p-CAH)_3_. 

Gd(III) complexes with p-CAH_2_ and CFAH_3_ are hydrated compounds. In the process of dehydration in the first stage, these complexes lose water of crystallization to about 140 °C. The Gd(CFAH_2_)_3_ loses four water molecules, while the Gd(p-CAH)_3_ loses two. In the further process of thermal decomposition, the complexes lose their coordination water, respectively, three molecules for (Gd(p-CAH)_3_) and four molecules for Gd(CFAH_2_)_3_, up to a temperature of about 250 °C. Above this temperature, thermal degradation of the complexes occurs. The final product of the thermal degradation of the complexes (about 900 °C) is Gd(III) oxide (Gd_2_O_3_). The thermal decomposition processes of Eu(III) complexes with p-CAH_2_ and CFAH_3_ proceed analogously. In the first stage of dehydration, the complexes lose crystallization water (Eu(CFAH_2_)_3_—three molecules, Eu(p-CAH)_3_—two molecules). This process runs up to a temperature of 120 °C. In the further stage of dehydration, the coordination water is lost. Eu(CFAH_2_)_3_ contains four coordination water molecules and Eu(p-CAH)_3_ contains three water molecules. Above 250 °C, thermal degradation of Eu(III) complexes occurs. The final decomposition product of both complexes is Eu(III) oxide (Eu_2_O_3_) (at a temperature of about 750 °C). In the case of decomposition of the caffeic acid complex, the final product contains the remains of organic carbon, which is burnt at a temperature above 750 °C. Among the tested complexes, the most thermally stable is the Gd(III) complex with CFAH_3_—its decomposition occurs above 250 °C. The least thermally stable is the Dy(III) complex with CFAH_3_. Its decomposition begins at about 200 °C.

### 2.2. Spectroscopic Study

The FTIR spectrum of p-CAH_2_ (Figure 3) showed the presence of a strong band originating from the stretching vibrations of the substituent in the aromatic ring ν(OH)_ar_ at a wavenumber of 3386 cm^−1^ (FTIR_KBr_) and a band assigned to the deforming in-plane vibrations of the β(OH)_ar_ at 1247 cm^−1^ (FTIR_KBr_) and 1259 cm^−1^ (FT-Raman) (Table 3). In the spectrum of p-CAH_2_, the characteristic bands originating from the stretching vibrations of the carbonyl group ν(C=O) at 1672 cm^−1^ (FTIR_KBr_), out-of-plane deformations of the γ(CO) at 919 cm^−1^ (FTIR_KBr_) and in-plane deformations of β(CO) at 692 cm^−1^ (FTIR_KBr_) and 678 cm^−1^ (FT-Raman) were observed. There was also a strong band originating from the stretching vibrations of the ν(C-OH) group near 1283 cm^−1^ (FTIR_KBr_) and 1281 cm^−1^ (FT-Raman). The presence of bands assigned to the stretching vibrations of the hydroxyl group ν(OH)_COOH_ in the wavenumber range of 2842–2578 cm^−1^ and deforming vibrations out-of-plane of γ(OH)_COOH_ at 557 cm^−1^ (FTIR_KBr_) were observed. In the range of wavenumbers 3192–2962 cm^−1^, there were bands originating from the C-H stretching vibrations of the aromatic ring and the aliphatic chain [33].

The attachment of a lanthanide ion (Eu(III), Dy(III), Gd(III)) to the p-CAH_2_ ligand causes characteristic changes in the FTIR and FT-Raman spectra of complexes compared with the spectrum of ligand. Among others, the disappearance of the bands assigned to the vibrations of the carbonyl group C=O and the hydroxyl group -OH: ν(OH)_COOH_, γ(OH)_COOH_, ν(C=O), γ(CO), β(CO) were observed. Chelation of the lanthanide ion to the p-CAH_2_ molecule resulted in the disappearance of the bands assigned to the ν(C-OH) stretching vibrations and the occurrence of the bands derived from the vibrations of the carboxylate anion (COO^−^) in the spectra of the complexes comparing with the spectrum of the ligand. In the spectra of complexes of p-CAH_2_, the stretching asymmetric vibrations ν_as_(COO^−^) occurred at the spectral range: 1513–1514 cm^−1^ (FTIR_KBr_) and 1519–1524 cm^−1^ (FT-Raman), symmetric stretches ν_s_(COO^−^) at 1409–1415 cm^−1^ (FTIR_KBr_) and 1401 cm^−1^ (FT-Raman), symmetric deforming in-plane vibrations β_s_(COO^−^) at 984 cm^−1^ (FTIR_KBr_) and 979–987 cm^−1^ (FT-Raman) and symmetric out-of-plane deformations γ_s_(COO^−^) in the range 732–735 cm^−1^ (FTIR_KBr_) and 726–733 cm^−1^ (FT-Raman).

The differences (Δν) between the wavenumbers of the bands assigned to the asymmetric and symmetric stretching vibrations of the carboxylate anion COO^−^ from the FTIR spectra of Dy(III), Eu(III), Gd(III) complexes and sodium salt of p-CAH_2_ were gathered in Table 4. The Δν values determined for the sodium salt of carboxylic acid and its lanthanide complexes provide information about the type of metal–ligand coordination [36]. When the difference between the wavenumbers of the asymmetric and symmetric stretches of the COO^−^ group Δν(COO^−^)complex from the IR spectra of the studied complex is much lower than the difference between wavenumbers of the same bands Δν(COO^−^)Na salt from the IR spectra of the sodium salt, the type of coordination is bidentate chelating. When Δν(COO^−^)complex ≤ Δν(COO^−^)Na salt, the bidentate bridging type of coordination is suspected in the studied complex. On the other hand, when Δν(COO^−^)complex ≫ Δν(COO^−^)Na salt, the metal ion in the complex is coordinated by the monodentate COO^−^ group.

Considering the above, bidentate chelating metal ion coordination in the studied lanthanide complexes with p-CAH_2_ was suggested.

The vibrational bands of the aromatic system present in the FTIR and FT-Raman spectra of the complexes had a similar intensity and location compared to the spectrum of p-CAH_2_. In the spectra of lanthanide complexes some of the aromatic bands were slightly shifted to the higher wavenumbers (i.e., bands marked with numbers 8a, 17a, 10a, 17b, 1, 16b) or to the lower wavenumbers (i.e., bands no. 8b, 19b, 9a) compared with the spectra of ligand. It suggests that the lanthanide ions did not significantly affect the aromatic system of p-CAH_2_. 

The spectra of CFAH_3_ (Figure 4) showed a strong band derived from the stretches ν(C=O) at 1645 cm^−1^ (FTIR_KBr_) and 1641 cm^−1^ (FT-Raman) as well as deforming in-plane vibrations β(C=O) at 817 cm^−1^ (FTIR_KBr_) and 648 cm^−1^ (FT-Raman), out-of-plane deformations γ(C=O) at 699 cm^−1^ (FTIR_KBr_) and 686 cm^−1^ (FT-Raman) and the stretching vibrations ν(C-OH) at 1281 cm^−1^ (FTIR_KBr_) (Table 5). The presence of bands of stretching vibrations of the hydroxyl group ν(OH)_COOH_ in the range of 2850–2573 cm^−1^ and deforming out-of-plane vibrations of γ(OH)_COOH_ at 590 cm^−1^ (FTIR_KBr_) were observed. In the spectrum of CFAH3, three bands corresponding to the vibrations of the substituent in the aromatic ring—hydroxyl group -OH were assigned: the stretching vibrations ν(OH)_ar_ at 3435 cm^−1^ and 3234 cm^−1^ (FTIR_KBr_), deforming in-plane vibrations of the β(OH)_ar_ 1297 cm^−1^ (FTIR_KBr_, FT-Raman). 

In the FTIR spectra of lanthanide complexes with CFAH_3_, the bands originating from the stretching vibrations of the C=O carbonyl group and the hydroxyl group -OH were not observed, whereas strong bands derived from the stretches of the carboxylate anion appeared at the following wavenumbers: asymmetric stretching vibrations ν_as_(COO^−^) 1499–1513 cm^−1^ (FTIR), symmetric stretching vibrations ν_s_(COO^−^) at 1401–1409 cm^−1^ (FTIR), symmetric in-plane deformations β(COO^−^) at 975–982 cm^−1^ (FTIR) and symmetric out-of-plane deformations γ_s_(COO^−^) plane at 695–731 cm^−1^ (FTIR). Similar to lanthanide complexes with p-CAH_2_, the wavenumbers of asymmetric and symmetric stretching vibrations of the carboxylate anion COO^−^ for CFAH_3_ complexes with Dy(III), Eu(III) and Gd(III) and the sodium salt have been gathered (Table 6). 

The FTIR spectrum of CFAH_3_ contained bands originating from the stretching vibrations of substituted -OH groups in the aromatic ring ν(OH)_ar_, located at 3435 and 3235 cm^−1^. In the spectra of lanthanide complexes with CFAH_3_, a significant shift of the first bands towards lower wavenumbers, i.e., 3418–3423 cm^−1^, and the disappearance of the second bands were observed. This indicates that besides forming of the carboxylate anion in the structures of lanthanide-CFAH_3_ complexes, the lanthanide ions were bonded by a hydroxyl group from the aromatic ring of the catechol moiety of CFAH_3_. The difference between the asymmetric and symmetric vibrations of the carboxylate anion corresponds to bidentate chelation coordination [36]. The analysis of the bands corresponding to the vibrations of the hydroxyl groups of the aromatic ring indicates, however, that these groups coordinate the metal, which was demonstrated and presented in our previous work [19]. The analysis of the IR spectra indicates that the lanthanide ions can coordinate the caffeic acid molecules through the carboxyl and hydroxyl groups.

Moreover, in the spectra of lanthanide- CFAH_3_ complexes, a decrease in the intensity of many bands assigned to the vibrations of the aromatic system and a shift towards lower wavenumbers were observed (bands no. 8a, 20b, 14, 18a, 17a, 10a, 6a, 16b, 8b). The effect of the lanthanide ions on the aromatic system of CFAH_3_ is stronger than in the case of p-CAH_2_, which results from a different type of metal–ligand coordination. The apparent shift of the aromatic band towards lower wavenumbers in the spectra of complexes compared to the spectra of CFAH_3_, as a result of a decrease in the strength of the C-C aromatic bonds, indicates that the lanthanide ions coordinated through the hydroxyl groups of the aromatic ring cause a decrease in the aromaticity and electronic stability of the aromatic ring of CFAH_3_.

Based on the FTIR and FT-Raman spectra of studied compounds and the elemental and thermogravimetric analysis results, the type of coordination modes of lanthanide complexes with p-CAH_2_ and CFAH_3_ were proposed (Figure 5). 

### 2.3. Antioxidant Activity

The antioxidant properties of Eu(IIII), Gd(III), and Dy(III) complexes of p-CAH_2_ and CFAH_3_ were studied using three different spectroscopic assays. The tested concentrations of compounds are given in Section 2.5. The molar metal to ligand ratio was 1:1, which is consistent with the results obtained in our previous work, where we investigated the complexing ability of CFAH_3_ and p-CAH_2_ towards Eu(III), Gd(III), and Dy(III) cations over a wide range of pH values [19,38]. Both assays, DPPH and ABTS, are recognized as a mixed HAT (hydrogen atom transfer) and SET (single electron transfer) mechanism that is generally called SPLET (sequential proton loss—electron transfer) or SET-PT (single electron transfer—proton transfer) mechanism [37]. A firm conclusion can be drawn regardless of the reaction mechanism between the antioxidant and the radical (in DPPH and ABTS tests) or Fe^2+^ ion (in FRAP assay). The Eu(IIII), Gd(III), and Dy(III) complexes of p-CAH_2_ showed higher antioxidant properties than the free ligand. On the contrary, the antioxidant activity of CFAH_3_ complexes with lanthanides was lower compared to the ligand. The obtained parameters for p-CAH_2_ were: IC_50_ = 13420.75 ± 664.28 µM, IC_50_ = 8.45 ± 0.14 µM, 263.10 ± 1.69 µM Fe^2+^, respectively, in DPPH, ABTS and FRAP assays. (Figure 6, Figure 7 and Figure 8) Whereas for CFAH_3_, the following data were obtained: IC_50_ = 6.46 ± 0.11 µM, IC_50_ = 4.13 ± 0.16 µM, 468.15 ± 2.84 µM Fe^2+^, respectively, in DPPH, ABTS, and FRAP assays. For the metal complexes of p-CAH_2_, the IC_50_ obtained in the DPPH test was in the range 7243.29–3948.92 µM, the IC_50_ obtained in the ABTS test: 5.42–4.07 µM; and in the FRAP assay: 292.97–267.68 µM Fe^2+^. In the case of metal complexes with CFAH_3_, the obtained IC_50_ parameters were similar and the FRAP values were lower compared with the data obtained for CFAH_3_ (i.e., the IC_50_ obtained in the DPPH test were in the range 7.27–6.18 µM; the IC_50_ obtained in the ABTS test: 5.42–4.07 µM; and in the FRAP assay: 452.76–432.68 µM Fe^2+^). CFAH_3_ possessed higher antioxidant activity than p-CAH_2_. This is caused by the presence of the -OH group in the ortho position in the aromatic ring of CFAH_3_. Complexation of CFAH_3_ with Eu(III), Gd(III), and Dy(III) did not enhance the antioxidant activity of CFAH_3_. On the contrary, the complexation of p-CAH_2_ with the same metal ions increased the antioxidant potential of this ligand (especially against DPPH^•^ radical and ABTS^•+^). 

What may be the reason for the differences in the effect of metal ions on the antioxidant activity of the ligands? As we presented earlier, based on spectroscopic studies, we found that the lanthanide complexes with CFAH_3_ and p-CAH_2_ differ in the type of metal-ligand coordination. In the case of p-CAH_2_, the carboxylate anion participates in the coordination of metal ions, whereas the coordination of metal ions in CFAH_3_ complexes takes place through the catechol moiety. The participation of the -OH from the aromatic ring in the coordination of metal ions reduces the hydrogen atom transfer from the antioxidant to the radical to some extent. Therefore, the antioxidant activity of metal complexes of CFAH_3_ is lower or similar to that of the ligand. On the other hand, in the coordination of metal ions through the carboxylate anion, there is a change in the electronic charge distribution in the ligand molecule [39,40], which favors the transfer of hydrogen atom from the antioxidant to the radical. These conclusions are also consistent with the results of our previous work [38].

Interestingly, a distinct relationship was noticed between the antioxidant activity of p-CAH_2_ and CFAH_3_ complexes and the type of coordinated ion. The highest antioxidant activity revealed Gd(III) complexes, which possessed the lowest IC_50_ values in the DPPH and ABTS assays and the highest FRAP value (in the case of CFAH_3_). Taking into account the increase in the antioxidant activity of studied metal complexes, they can be ordered as follows: Dy(III) < Eu(III) < Gd(III) complexes.

### 2.4. Antimicrobial Activity

The results indicate more substantial antibacterial and antifungal effects of the complexes of CFAH_3_ and p-CAH_2_ compared to pure acids (Table 7, Table 8 and Table 9). *C. albicans* shows the highest resistance to all tested compounds. This fact is probably due to the high ability of this microorganism to form biofilms that reduce the availability of the antifungal agent and the high frequency of spontaneous mutations and overexpression of multidrug efflux pumps [41,42,43]. Gram-positive and Gram-negative bacteria show a similar level of resistance to the tested compounds. Metal-based antibacterial compounds are very effective against more resistant Gram-negative bacteria due to their multifaceted action. Most metal ions have been shown to disrupt the cell walls of Gram-negative bacteria. In addition, metal ions easily penetrate the cell walls of both Gram-positive and Gram-negative bacteria. All this makes them excellent candidates for developing new antibiotics [42]. Due to the ever-increasing antibiotic resistance among bacteria, especially Gram-negative bacteria, and the high toxicity of gentamicin, there is a great need to search for new bactericidal compounds. Metal-based complexes seem to have great potential for this, although they often have an antibacterial effect in much higher concentrations than classic antibiotics. The results indicate that the greatest antibacterial and antifungal potentials were exhibited by complexes of CFAH_3_ and p-CAH_2_ with Eu(III) and Gd(III). Differences in MIC values, compared to the results presented in our previous work for complexes of Eu(III) with CFAH_3_ [19], may be due to species-specific differences in the cell metabolism of the tested microorganisms and thus antimicrobial susceptibility. For example, *B. subtilis and E. coli* perform a fermentation process that can significantly reduce the organic compound level of the medium [44]. In addition, bacteria are very susceptible to various mutations, which may result in changes in the activity of membrane protein carriers, which results in less penetration of organic compounds into the cells. Over the years, much knowledge has been accumulated on organic compounds’ pharmacological and metabolic effects. In the case of metal complexes, this is still unexplored. Despite the advantage of organic compounds as antibiotics, metals have some unique benefits. Metal complexes have several unique mechanisms of action. They can undergo ligand exchange reactions to release bioactive molecules, which is associated with higher antimicrobial potential [45].

### 2.5. Microelectrophoretic Mobility Measurements

Surface charge (*δ*) and zeta potential (*ζ*) indicate changes in cell envelope compositions occurring during the interaction of bacteria or fungi with ions and particles. Since natural membranes are complex assemblies of molecules, their study is highly engaging. Therefore, the properties of natural membranes are often studied using liposomes.

We formed liposomes representing three different types of microorganism membranes, including those of bacterial and fungal cells. The major anionic and zwitterionic phospholipid species found in Gram-negative and Gram-positive bacterial cells are palmitoylphosphatidylglycerol (POPG) and palmitoylphosphatidylethanolamine (POPE). To model the surfaces of Gram-positive bacteria, such as *Bacillus subtilis* [46], a 3:1 ratio of POPG/POPE was simulated. However, to model the surfaces of Gram-negative bacteria, such as *Escherichia coli*, purchased *E. coli* polar lipid extract was used, which besides PG and PE, additionally contained the double-negatively charged lipid cardiolipin, being a minor constituent (representing ~5–10%) of the cytoplasmic *E. coli* membrane [47]. The model of the cell surface of fungi presented in this article was based on the cell membrane of baker’s yeast (*Saccharomyces cerevisiae*), an organism often used in biomolecular studies owing to the relative simplicity of its lipidome and its uncomplicated growth requirements [48]. This model included the zwitterionic phospholipid DPPC possessing a neutral charge at the interface, and the saturation of the lipid enabled dense packing of the molecules. The dipalmitoyl chains formed the most significant proportion of the saturated lipid fatty acids in yeast, constituting up to 20% of the total lipids, with the remainder being unsaturated [49]. The second phospholipid in the simple bilayer model, possessing a negative charge, was PS, which belongs to the yeast cell membrane’s most abundant anionic phospholipid group [50]. Phosphatidylcholines and phosphatidylserines are also found in high proportions in the cell membranes of other fungi, for example, selected strains of the human pathogen *Candida albicans* [51]. 

The values of the surface charge and zeta potential of liposomes formed from phospholipids prevalent in cell surfaces of fungi, Gram-positive and Gram-negative bacteria, as a function of pH in the 0.2 mol⋅dm^−3^ KCl_(aq)_ solutions containing ligand (CFAH_3_ or p-CAH_2_), Dy or Dy with phenolic acid at 1:1 the molar metal to ligand ratio (*c =* 10^−3^ mol⋅dm^−3^), are plotted in Figure 9. Briefly, liposomes placed in the 0.2 mol⋅dm^−3^ KCl_(aq)_ were treated as a control group, and their *δ* and *ζ* were relatively constant from either pH 4.0 (artificial surfaces of Gram-positive bacteria and fungi) or 5.0 (artificial surfaces of Gram-negative bacteria) to 10.0. However, at pH = 2 and pH = 3, the values of both parameters markedly increased due to negative membrane-surface charge neutralization by the excess of protons in the surrounding environment. It can be noticed that liposomes mimicking Gram-negative bacteria cells, due to the higher amount of anionic phospholipids included in their composition, exhibited a higher negative surface charge and a higher negative potential than Gram-positive bacteria. The trends in these results are noticeably similar to those obtained for natural microorganism cells due to the presence of the negatively charged lipopolysaccharides layer in the outer membrane of Gram-negative bacteria, contrary to Gram-positive bacteria [19].

CFAH_3_ and p-CAH_2_ did not affect *δ* and *ζ* of liposomal membranes when measurements were performed for pH 2.0 and 3.0 (excluding p-CAH_2_ and liposomes modeling of the surface of Gram-positive bacteria, Figure 10b). At acidic pH, these compounds can dissolve and permeate the membrane. However, in neutral and alkaline pH, the carboxylic acid group of phenolic acid anchors it to the polar heads of membrane phospholipids. Our results confirm this hypothesis for liposomes mimicking the Gram-positive bacteria surface, showing more negative *δ* and *ζ* values with the addition of the CFAH_3_ or p-CAH_2_. 

In the presence of the Dy(III) or Dy(III)/ligand complex, surface charge and zeta potential were moved to neutral values due to interaction at the interface leading to increased surface tension. It has been documented that the increased surface tension and zeta potential can significantly affect the surfaces of microorganisms. As a result of these phenomena, the cell surface shows some abnormal textures such as rupture, blebs, etc. In addition, the ruptured cells no longer remain intact and form aggregates or clumps [52].

The pH dependencies of the surface charge and the zeta potential of liposomes mimicking microbial surfaces formed in an electrolyte solution (0.2 mol⋅dm^−3^ KCl_(aq)_) containing Gd(CFAH_2_)_3_ or Gd(p-CAH)_2_ complexes demonstrate the same trend as those determined in the presence of the Dy(III) complexes. In contrast, the dependencies obtained in the presence of CFAH_3_ and p-CAH_2_ complexes with Eu(III) differ significantly from those obtained for Dy(III) and Gd(III) and their complexes with phenolic acids and are shown in Figure 10.

Generally, a remarkable value of zeta potential, greater than 20 mV (positive or negative), is adopted as the arbitrary value that should stabilize the liposomes and protect against their aggregation [52]. Thus, the zeta potential might be a valuable source of information, shedding additional light on liposome properties. For example, from the data shown in Figure 9 and Figure 10, one may conclude that liposomes mimicking microbial surfaces formed in the electrolyte containing phenolic acid were significantly more stable than liposomes formed in the electrolyte containing lanthanide ions or lanthanide ions complexes with phenolic acid.

The antibacterial activity of caffeic acid and its derivatives towards bacterial and fungal pathogens is known mainly by changing the membrane permeability, inhibiting enzyme activity, and damaging protein structure and DNA [53]. Our results demonstrate that the most significant changes in zeta potential and surface charge occurred in the presence of Eu(III) and its complexes with p-CAH_2_ on the surface of liposomes modeling *E. coli* cell wall (Figure 10d). The obtained data are consistent with the antimicrobial activity results and demonstrate that both electrical parameters can be successfully used to interpret the antibacterial properties of lanthanide complexes with p-CAH_2_ and CFAH_3_.

## 3. Materials and Methods

### 3.1. Materials 

Caffeic acid (3-(3,4-dihydroxyphenyl)-2-propenoic acid, CFAH_3_), *p*-coumaric acid (4-hydroxycinnamic acid, p-CAH_2_), Eu(III) chloride hexahydrate (EuCl_3_·6H_2_O), Dy(III) chloride hexahydrate (DyCl_3_·6H_2_O), and Gd(III) chloride hexahydrate (GdCl_3_·6H_2_O), DPPH (2,2-diphenyl-1-picrylhydrazyl), ABTS (2,2-azino-bis(3-ethylbenzothiazoline-6-sulfonic acid), potassium persulfate (K_2_S_2_O_8_), TPZT (2,4,6-tris(2-pyridyl)-s-triazine), iron(III) chloride hexahydrate (FeCl_3_∙6H_2_O), iron(II) sulfate (FeSO_4_), potassium bromide (KBr), gentamycin and fluconazole were purchased from Sigma-Aldrich Co. (St. Louis, MO, USA). Phospholipids, namely 1-palmitoyl-2-oleoyl-*sn*-glycero-3-phospho-(1′-*rac*-glycerol) (sodium salt) (16:0–18:1 PG, POPG, purity > 99%), 1-palmitoyl-2-oleoyl-*sn*-glycero-3-phosphoethanolamine (16:0–18:1 PE, POPE, purity > 99%), 1,2-diacyl-*sn*-glycero-3-phospho-L-serine (PS, purity > 97%), dipalmitoylphosphatidylcholine (16:0 PC, DPPC, purity > 99%), and *E. coli* polar lipid (EPL) extract, were also purchased from Sigma-Aldrich (St. Louis, MO, USA). Sodium hydroxide (NaOH) was purchased from POCH S.A. (Gliwice, Poland). Methanol was sourced from Merck (Darmstadt, Germany). Mueller-Hinton agar was supplied by Oxoid (Hampshire, UK). All chemicals had an analytical purity and were used without further purification. 

### 3.2. Synthesis

Eu(III), Dy(III), and Gd(III) complexes with p-CAH_2_ were obtained in a solid state. First, the weight amount of p-CAH_2_ (0.0001 mol) was dissolved in an appropriate volume of an aqueous solution of sodium hydroxide (0.1 mol/L), in a stoichiometric molar ratio *p*-CAH_2_:NaOH of 1:1, in a water bath at 50 °C assisted by ultrasounds. Then, aqueous solutions of Eu(III), Dy(III), and Gd(III) chlorides (0.01 mol/L) were prepared. The solutions of lanthanide chlorides and the previously prepared sodium salt of p-CAH_2_ were slowly mixed in a stoichiometric molar ratio of 1:3 (lanthanide chloride:sodium salt of p-CAH_2_). The mixture was shaken on a shaker at room temperature for 1 h. The cloudy solutions were allowed to precipitate for 3 days. The precipitates were then filtered off and left at room temperature until residual water evaporated. The complexes prepared in this way were stored in a desiccator. Elemental analysis showed that the hydrated complexes of Eu(III), Dy(III), and Gd(III) with p-CAH_2_ were obtained, and the molar metal/ligand ratio was 1:3.

Complexes of Eu(III), Dy(III), and Gd(III) with CFAH_3_ were also obtained in the form of solids. In the first stage, an appropriate mass of CFAH_3_ (0.0001 mol) was dissolved in an aqueous solution of sodium hydroxide (0.1 mol/L) in a stoichiometric molar CFAH_3_:NaOH ratio of 1:1. The process was conducted in a water bath at 50 °C assisted by ultrasounds. A yellow solution was formed, which was then slowly added (with stirring) to a 0.01 mol/L solution of Dy(III) chloride in a molar ratio of 1:3 (Dy (III) chloride/sodium salt of CFAH_3_). The mixture was shaken on a shaker at room temperature for 1 h. The cloudy solutions were allowed to precipitate for 3 days. The precipitates were then filtered off and left at room temperature until residual water evaporated. The analogous procedure was conducted to prepare Eu(III) and Gd(III) complexes with CFAH_3_. The mixtures were shaken on a shaker at room temperature for 2 h. Then, the same procedure was undertaken as in the case of complexes with p-CAH_2_. Elemental analysis showed that hydrated complexes were obtained, and the molar metal to ligand ratio was 1:3.

### 3.3. Elemental and Thermal Analysis

Thermal analysis was performed on a STA 600 Frontier thermal analyzer (Perkin Elmer, Waltham, MA, USA). Samples (5–6 mg) were annealed in a ceramic crucible in the temperature range of 40–995 °C in an air atmosphere at a heating rate of 10 °C/min. Thermal decomposition products were analyzed based on the recorded TG, DTG, and DSC curves. Elemental analysis was performed on a Perkin Elmer Series Analyzer II CHNS.

### 3.4. Raman and FTIR Spectroscopy

FTIR spectra were recorded using the ATR technique (FTIR_ATR_) and the method of pressing KBr (FTIR_KBr_) pellets in the range of 400–4000 cm^−1^ on an Alfa spectrometer (Bruker, Billerica, MA, USA). FT-Raman spectra were recorded on a MultiRam apparatus (Bruker, Billerica, MA, USA) in the range of 400–4000 cm^−1^ at a laser power of 200 mW.

### 3.5. Antioxidant properties

The antioxidant assays (DPPH, ABTS, and FRAP) were performed according to the methodology described in the literature [21]. Absorbance measurements were carried out using the UV-VIS Carry 5000 spectrophotometer (Santa Clara, CA, USA) and Tecan Infinite 200 PRO microplate reader (Tecan, Männedorf, Switzerland). Solutions of Eu(III), Dy(III), and Gd(III) complexes with CFAH_3_ and p-CAH_2_ were prepared with a metal-ligand composition of 1:1, following the results of previous studies on the composition of lanthanide complexes with CFAH_3_ and p-CAH_2_ depending on pH [19,38]. First, an 10% methanolic CFAH_3_ (0.001 and 0.0001 mol/L) and p-CAH_2_ (0.05 and 0.005 mol/L) solutions were added to an aqueous solution of sodium hydroxide (0.1 mol/L) in a stoichiometric molar ligand/NaOH ratio of 1:1. Then, aqueous solutions of Eu(III), Dy(III), and Gd(III) chlorides (0.01 mol/L) were prepared and mixed with sodium salts of ligands so that the molar ratio of lanthanide chloride/sodium salt of the ligand was 1:1. The final concentration of metal complexes for DPPH determination was in the range of 2.67–12.00 mmol/L (for CFAH3 and its complexes) and 2.5–23.3 mmol/L (for p-CAH_2_ and its complexes), for ABTS assay 2.4–8.6 µmol/L (for CFAH_3_ and its complexes) and 1.0–16.0 µmol/L (for p-CAH_2_ and its complexes), and for FRAP assay—11.76 µmol/L (for all tested compounds). All experiments were carried out in five repetitions for three independent experiments, and values were expressed as mean ± SD.

### 3.6. Antimicrobial Study

The MIC was determined by serial dilution of Eu(III), Dy(III), and Gd(III) complexes with CFAH_3_ and p-CAH_2_ (where the molar metal/ligand ratio was 1:1) in an agar medium to which the appropriate inoculum of microorganisms was then added and incubated. The microorganisms used in this study came from the Polish Collection of Microorganisms (Wroclaw, Poland): *E. coli* (PCM 2857), *B. subtilis* (PCM 2850), and *C. albicans* (PCM 2566-FY). Bacteria were grown overnight and then re-suspended in physiological saline to an optical density at 600 nm (OD600) of 0.60, corresponding to 5.0 × 10^8^ CFU/mL. Bacteria (0.1 mL of reconstituted suspension) were seeded on sterile Mueller–Hinton agar (Oxoid) plates, to which appropriate amounts of tested compounds were added previously to obtain the desired concentration. Tested chemicals were diluted in a DMSO solution (Sigma-Aldrich**)**. Negative controls were agar plates, to which DMSO was added, and positive control were plates with gentamicin (Sigma-Aldrich) (in the case of bacteria) or fluconazole (Sigma-Aldrich) (in the case of fungi). The plates were incubated at 37 °C for 24 h. The lowest concentration without visible bacterial growth was determined as MIC (Minimal Inhibitory Concentration). 

### 3.7. Microelectrophoretic Mobility Measurements 

The surface charge and zeta potential of liposomes mimicking surfaces of the bacteria and fungi cells were determined by performing micro-electrophoretic assessments on samples using the Electrophoretic Light Scattering technique. The measurements were conducted with the Zetasizer Nano ZS apparatus (Malvern Instruments, Malvern, UK). The experiment was carried out as a function of pH using a WTW InoLab pH 720 laboratory meter (WTW, Weinheim, Germany). 

Liposomes were prepared according to the method proposed by Huang [54]. Firstly, 10 mg of phospholipid (POPG, POPE, DPPC, PS, or EPL) was dissolved in 1–2 mL of chloroform (anhydrous, ≥99%). The obtained solutions were mixed in appropriate molar ratios to achieve the desired systems modeling the cell surfaces of the microorganisms. Next, the solvent was evaporated under a gentle stream of argon. The dried lipid film was hydrated with an electrolyte solution (0.2 mol⋅dm^−3^ KCl_(aq)_). The last stage of liposome formation was sonication of the suspension five times for 90 s, each time using an ultrasound generator UD 20 (Techpan, Poland). 

The liposomes were suspended in the tested solutions and allowed to interact with the solution components for 1 h at room temperature. Additionally, 0.2 mol⋅dm^−3^ KCl_(aq)_ solutions containing free ligands (CFAH_3_ or p-CAH_2_), Ln(III) cations (Eu(III), Gd(III) or Dy(III)), or metal–ligand system at 1:1 ratio, at ca. 10^−3^ mol⋅dm^−3^ were used. Suspended liposomes were then titrated to the desired pH (range of 2–10) with strong acid (HCl) and strong base (NaOH) solutions, which were prepared with sodium chloride to keep the strength of the ionic solution constant. Six measurements were made (each covering 100–200 series for a duration of 5 s) for each pH value for each sample. The experiments were repeated three times with similar results.

The zeta potential values were calculated from the electrophoretic mobility using the Henry correction of Smoluchowski’s equation [55]:(1)$$\zeta =\frac{3\cdot \eta \cdot \mu }{2\cdot \varepsilon \cdot \varepsilon_{0}\cdot f(\kappa a)},$$
where: µ—the electrophoretic mobility, η—the viscosity of the aqueous solution, ε—the relative permittivity of the medium, ε_0_—the permittivity of free space, and ƒ(κa)—Henry’s function.

The surface charge values were calculated from the electrophoretic mobility using the equation [40]:(2)$$\delta =\frac{\eta \cdot µ}{d},$$
in which *d*—the diffuse layer thickness.

The diffuse layer thickness was determined from the formula [56]
(3)$$d=\sqrt{\frac{\varepsilon \cdot \varepsilon_{0}\cdot R\cdot T}{2\cdot F^{2}\cdot I}}$$
where, R—the gas constant, T—the temperature, *F*—the Faraday constant, I—the ionic strength of the electrolyte. 

## 4. Conclusions

The general formula of solid Eu(III), Dy(III) and Gd(III) complexes with p-CAH_2_ is [Ln(C_9_H_7_O_3_)_3_∙mH_2_O]∙nH_2_O, whereas with CFAH_3_ [Ln(C_9_H_7_O_4_)_3_∙nH_2_O]∙mH_2_O (where Ln—lanthanide ion; ‘n’ and ‘m’ are numbers of coordinated water which are in the range of 2–4). The composition of solid complexes was established based on elemental, thermal, and spectroscopic studies. In the complexes of p-CAH_2_, the metal ions were coordinated by the bidentate chelating carboxylate groups. In contrast, in the complexes of CFAH3, the carboxylic group is deprotonated as well, but the catechol moiety coordinated lanthanide ions. The obtained data suggested that lanthanide complexes are much more thermally stable and possess higher antimicrobial and antioxidant properties than the ligands alone (except CFAH_3_ in the case of antioxidant activity tests). Gd(III) complexes revealed the highest biological activity among the studied lanthanide complexes. 

## Data Availability

The datasets used and/or analyzed during this study are available from the corresponding author upon reasonable request.

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
