# Peer review of "Structures, Antioxidant Properties, and Antimicrobial Properties of Eu(III), Gd(III), and Dy(III) Caffeinates and p-Coumarates"

_molecules, 2023, doi:10.3390/molecules28186506_

Round 1

Reviewer 1 Report

The ms ‘’ Structures, antioxidant properties and antimicrobial study of europium(III), gadolinium(III), and dysprosium(III) caffeinates 3 and p-coumarates’’ by Grzegorz Åšwiderski et.al describes the synthesis and spectroscopic characterisation of Ln(III) [Eu, Dy, Gd] complexes with p-coumaric acid and caffeic acid as ligands. The antimicrobial and antioxidant properties of these complexes were also evaluated and described towards the properties of the free ligands and as a function of the binding modes of the two ligands. In this ms. the authors claim that the different coordination modes of the ligands with Eu, Dy and Gd lead to different antioxidant properties with the p-coumaric acid complexes (bidentate chelating mode) owing better performance towards the caffeic acid complexes (OH group chelating mode). The coordination modes of the two ligands in the Ln(III) complexes is only based on spectroscopic data. For p-coumaric acid complexes the IR data suggests the chelating mode of the acetato group of the ligand but for the caffeic acid the IR data are confusing. I believe that the authors are based on speculation about the coordination modes and the types of complexes. On the contrary, the antioxidant and antimicrobial properties of these complexes point out that the speculations were correct. Overall, the work is organized well but the speculations for the types of the complexes and the coordination modes of the ligands should be addressed before publication. I believe that this work can be further improved after major revision.

Some points of consideration that supports my decision are:

·         The title of the ms should be changed and include Eu(III), Dy(III) and Gd(III) instead of europium (III) etc.

·         In the main text it is better to use the element symbols instead of the full names of the elements, e.g. Eu(III) instead of europium.

·         The meaning of the paragraphs in lines 50-53 should be revised to make more sense.

·         Ligands p-CA and CFA should be abbreviated as p-CAH2 and CFAH3 to denote the potential 3 acid H that can be deprotonated.

·         The complexes should be abbreviated in brackets and the ligands should be written in the form of the deprotonated form e.g. [Dy(p-CAH)3(H2O)2]∙3H2O

·         The IR peak numbers in Figures 3 and 4 should contain the numbers without the decimal. Also, there are minor discrepancies between peak numbers written in the text and shown in the Figures.

·         Tables 3 and 5 contain all the spectroscopic data should be placed in the SI as they only show the data presented in Figures 3 and 5.

·         The Raman Spectra are missing.

·         About the Nakamoto criteria that is also cited on reference 36, there is no such criteria. The Δ value parameter that indicated if the acetate group chelates or acts as a monodentate ligand has been described in the review of G.B. Deacon and R.J. Philips. Coordination Chemistry Reviews, 1980, 33, 227-250 (DOI: 10.1016/S0010-8545(00)80455-5). This reference should be placed along with Nakamoto’s handbook in ref. 36.

·         The coordination mode of the CFA ligand as mentioned above is supported only by spectroscopic data and these data are little confusing to me. I would suggest more experimental data such as single crystal X-ray crystallography, UV-Vis, pH titrations, etc. to support the proposed coordination mode.

·         The proposed types of the complexes are not charge balanced and again are based on speculations derived from IR spectra.

The quality of the English language used and the terminology about the coordination chemistry can be improved.

Author Response

Dear Reviewer,
thank you for your suggestions that helped to improve our manuscript. A list of responses (changes made to the manuscript) is attached.

Reviewer 2 Report

The main disadvantage of the proposed manuscript is the lack of chemistry. Not every reader knows what p-coumaric and caffeic acids are. Therefore, their structural formulas and systematic names at the beginning of the manuscript are necessary.

There are also big questions about the composition and structure of the resulting complexes. For caffeic acid complexes, a composition of 1:2 is assumed (see Table 1 and others), but this means either the oxidation state of the metal is 2+, which is impossible for gadolinium and dysprosium, or double deprotonation of one of the ligands. Moreover, proposed by the authors in Fig. 5, the model of coordination with a free carboxylate group clearly contradicts the chemical nature. The authors should think about the structure of the complexes, which satisfies the data obtained.

For a more complete characterization of the complexes, data on the metal content are needed.

The authors rightly write about the high toxicity of lanthanide salts. Therefore, for medical use, it is necessary to assess the stability of the synthesized complexes.

Author Response

(The authors gave the same response as above.)

Reviewer 3 Report

The authors have published a preprint with the URL and DOI number I have given below, which is exactly the same as the manuscript I have evaluated. I don't know if this poses a problem for the journal. 

Although such a situation does not pose a problem for the journal, the authors should have stated it somehow.

https://papers.ssrn.com/sol3/papers.cfm?abstract_id=4467299

https://dx.doi.org/10.2139/ssrn.4467299

My comments are valid if the situation I mentioned above is not a problem for the journal. Otherwise, I recommend rejecting the study.

In the manuscript entitled ‘’Structures, antioxidant properties and antimicrobial study of europium(III), gadolinium(III), and dysprosium(III) caffeinate and p-coumarates’’ the authors prepared some lanthanide complexes and fully characterized their structures with several analytical techniques. Also, they evaluated their biological activities. When looking at the previous studies of the authors, it is seen that they published many publications on lanthanides, and they published these publications in journals with high impact factors. Therefore, it can be considered that the authors are experts in the synthesis and structure elucidation of lanthanide complexes. In this study, unlike the others, they synthesized complexes of lanthanides such as europium(III), gadolinium(III), and dysprosium(III) and evaluated their antimicrobial and antioxidant properties.

When looking at other studies published in the journal, it is seen that the results presented in this study are suitable for the journal. I recommend that the publication could be accepted after the revisions below are made.

- Since the abstract is the first impression, it should be stated the activity results obtained in the abstract section of the study, especially whether higher activity was obtained compared to the reference.

- Whole the manuscript should be reviewed, and spelling errors should be checked. For example, the term ‘’Candida’’ is misspelled in the abstract section.

Minor editing of English required.

Author Response

(The authors gave the same response as above.)

Round 2

Reviewer 1 Report

The ms ‘’Structures, antioxidant properties, and antimicrobial study of Eu(III), Gd(III), and Dy(III) caffeinates and p-coumarates’’ has been significantly improved since the last time it was reviewed. The authors addressed all the issues raised by the reviewers, and thus I believe that now it is ready to be accepted for publication. I suggest the authors fully review the entire manuscript for spelling mistakes and corrections, as there are many scattered in the text (e.g., lines 33, 34, 54, 158, 302, 423). Also, in line 54, a scheme or figure caption should be added. As the types of the complexes and coordination modes are only suggested and proposed by spectroscopic data, I suggest the authors check the types of the complexes and, if there are discrepancies with the charge balance, leave only the ratio between metal and ligands. Also, the section on organometallic chemistry is not well suited for this ms. 

The quality of the English can be improved.

Author Response

Responce to Reviewer 1.

Answers:

The ms ‘’Structures, antioxidant properties, and antimicrobial study of Eu(III), Gd(III), and Dy(III) caffeinates and p-coumarates’’ has been significantly improved since the last time it was reviewed. The authors addressed all the issues raised by the reviewers, and thus I believe that now it is ready to be accepted for publication. I suggest the authors fully review the entire manuscript for spelling mistakes and corrections, as there are many scattered in the text (e.g., lines 33, 34, 54, 158, 302, 423). Also, in line 54, a scheme or figure caption should be added. As the types of the complexes and coordination modes are only suggested and proposed by spectroscopic data, I suggest the authors check the types of the complexes and, if there are discrepancies with the charge balance, leave only the ratio between metal and ligands. Also, the section on organometallic chemistry is not well suited for this ms

Authors: Thank you for your review. We made corrections. Language errors have been corrected.

Reviewer 2 Report

Carrying out any biological studies of new compounds makes sense only if there is reliable information about their composition and structure. Otherwise, such studies do not provide any useful information about the structure-property relationship. The identification of new compounds obtained by the authors is based on the interpretation of data from 3 different experiments, namely, chemical analysis, thermogravimetry, and IR spectroscopy. At the same time, the latter does not provide information on the composition of the compounds, but allows one to make some assumptions about their structure.

Chemical analysis data are based on the determination of the content of two elements (carbon and hydrogen), which total less than 50 wt. % and therefore are not a reliable indicator. Therefore, in this case, a good correlation with thermograviometric data is very important. In the initial version of the manuscript, the authors did not quite correctly interpret the chemical analysis data and proposed an erroneous formula for one of the series of compounds (caffeic acid complexes).

In the revised version of the manuscript, the authors present updated chemical analysis data, on the basis of which a different formula for the same series of compounds was proposed. However, in this case, this does not correlate well with the thermogravimetry data. It is not very clear to me how different results about the water content can be obtained from the same thermograviometric curves. In addition, the total weight loss found for these complexes is very different from the calculated one (for example, 79.9 wt. % versus 66.8 wt. % for the europium complex). This raises serious doubts about the reliability of the interpretation given by the authors of the initial data for these compounds regarding their composition and formula. Considering these data to be very unreliable, I would strongly recommend that the authors remove this series of compounds from the manuscript.

I would also like to comment on the description of the synthesis of complexes in the experimental part. It does not contain any details of the experiment that would allow an independent researcher to reliably reproduce the results. In addition, there are no data on the yields of the synthesized complexes. This is unacceptable and should be corrected.

Author Response

Responce to reviewer 2

Answers:

  • Carrying out any biological studies of new compounds makes sense only if there is reliable information about their composition and structure. Otherwise, such studies do not provide any useful information about the structure-property relationship. The identification of new compounds obtained by the authors is based on the interpretation of data from 3 different experiments, namely, chemical analysis, thermogravimetry, and IR spectroscopy. At the same time, the latter does not provide information on the composition of the compounds, but allows one to make some assumptions about their structure.

Authors:

The tests of the obtained compounds in the solid phase were only supplementary, the composition of the analyzed compounds in the aqueous phase is important. In this form, the compounds are subjected to biological tests. As already mentioned, these types of complexes (their composition) were studied in detail in the aqueous phase in our previous works.

  • Chemical analysis data are based on the determination of the content of two elements (carbon and hydrogen), which total less than 50 wt. % and therefore are not a reliable indicator. Therefore, in this case, a good correlation with thermograviometric data is very important. In the initial version of the manuscript, the authors did not quite correctly interpret the chemical analysis data and proposed an erroneous formula for one of the series of compounds (caffeic acid complexes).In the revised version of the manuscript, the authors present updated chemical analysis data, on the basis of which a different formula for the same series of compounds was proposed. However, in this case, this does not correlate well with the thermogravimetry data. It is not very clear to me how different results about the water content can be obtained from the same thermograviometric curves. In addition, the total weight loss found for these complexes is very different from the calculated one (for example,9 wt. % versus 66.8 wt. % for the europium complex). This raises serious doubts about the reliability of the interpretation given by the authors of the initial data for these compounds regarding their composition and formula. Considering these data to be very unreliable, I would strongly recommend that the authors remove this series of compounds from the manuscript.

Authors:

Thermal data were reinterpreted and compared with elemental analysis. The authors improved the descriptions of the thermal analysis results. The differences between the calculated and the experimental content of residues result from the fact that the organic carbon was not completely burnt out in the range of the temperatures used. The chart shows a further decrease in the TG curve.

  • I would also like to comment on the description of the synthesis of complexes in the experimental part. It does not contain any details of the experiment that would allow an independent researcher to reliably reproduce the results. In addition, there are no data on the yields of the synthesized complexes. This is unacceptable and should be corrected.

Authors: The synthesis of the compounds is described in Section 3.2 Synthesis. The results of the synthesis yield have been supplemented.

Reviewer 3 Report

The authors have obviously made the required changes after reviewing the revised draft. The revisions are sufficient and convincing for me. When looking at other studies published in the journal, it is seen that the results presented in this study are suitable for the journal. I recommend that the publication could be accepted.

Author Response

Responce to reviewer 3

Answers:

The authors have obviously made the required changes after reviewing the revised draft. The revisions are sufficient and convincing for me. When looking at other studies published in the journal, it is seen that the results presented in this study are suitable for the journal. I recommend that the publication could be accepted.

Authors: Thank you for your review.